# How to Increase Cellular Glutathione

**DOI:** 10.3390/antiox12051094

**Published:** 2023-05-13

**Authors:** Daniela Giustarini, Aldo Milzani, Isabella Dalle-Donne, Ranieri Rossi

**Affiliations:** 1Department of Biotechnology, Chemistry and Pharmacy, University of Siena, 53100 Siena, Italy; daniela.giustarini@unisi.it; 2Department of Biosciences, University of Milan, 20133 Milan, Italy; aldo.milzani@unimi.it (A.M.); isabella.dalledonne@unimi.it (I.D.-D.)

**Keywords:** glutathione, oxidative stress, glutathione boosters, cysteine prodrug, Nrf2 activators, N-acetyl cysteine

## Abstract

Glutathione (GSH) has special antioxidant properties due to its high intracellular concentration, ubiquity, and high reactivity towards electrophiles of the sulfhydryl group of its cysteine moiety. In most diseases where oxidative stress is thought to play a pathogenic role, GSH concentration is significantly reduced, making cells more susceptible to oxidative damage. Therefore, there is a growing interest in determining the best method(s) to increase cellular glutathione for both disease prevention and treatment. This review summarizes the major strategies for successfully increasing cellular GSH stores. These include GSH itself, its derivatives, NRf-2 activators, cysteine prodrugs, foods, and special diets. The possible mechanisms by which these molecules can act as GSH boosters, their related pharmacokinetic issues, and their advantages and disadvantages are discussed.

## 1. Introduction

Glutathione (GSH) is a tripeptide (γGlu-Cys-Gly) that is the major low molecular mass thiol in mammals. It is characterized by a gamma-peptide bond between the carboxyl group of the glutamate side chain and cysteine (Figure 1). This unusual amide bond protects the molecule from degradation by the action of cellular proteases. The carboxyl group of the cysteine residue is attached to glycine via a normal peptide bond. From a chemical point of view, this tripeptide has one positively charged amino group and two negatively charged carboxyl groups at a physiological pH. These functional groups make GSH an extremely hydrophilic molecule, capable of interacting with various macromolecules through electrostatic attraction, also through hydrogen bonds.

GSH occurs in millimolar concentrations in cells, where it participates in detoxification reactions of both oxidants and electrophilic compounds. Therefore, the maintenance of physiological GSH levels is crucial to protect cells from harmful molecules [1].

It has been reported that various diseases with known or unknown etiology have decreased GSH levels. This phenomenon is thought to be due to the inability of cells to restore normal GSH levels or to increased production of reactive oxygen species (ROS), which can damage macromolecules fundamental to cell survival before being deactivated by antioxidant defenses. In this sense, it is evident that strategies aimed at restoring GSH levels could represent a new therapeutic option to prevent or alleviate the progression of diseases caused by oxidative stress. Numerous preclinical studies have shown that increasing GSH levels is possible, but the clinical implications of these results are still negligible [2]. In other words, currently, almost any disease seems to benefit from treatment with molecules capable of increasing GSH in vivo. This may depend on various factors, including the fact that several molecules are only able to increase GSH only under extreme conditions (i.e., very high concentrations) and in in vitro models but are unable to do the same in vivo. 

The purpose of this review is to analyze what is reported in the literature in this field, focusing on the molecules and/or strategies that are more suitable to be used as GSH boosters in vivo. The molecular mechanisms involved (if known) will also be described.

## 2. Reactions of GSH 

The most important chemical functionality of GSH is its cysteinyl moiety, which can act as a radical scavenger and reducing agent via its sulfhydryl group (SH) or participate in detoxification reactions through its addition to the electrophilic center of xenobiotics [1]. Although the pKa value of Cys-thiol GSH is relatively high (8.9–9.4) [3], indicating low reactivity in the intracellular environment, its central role in protecting cells from reactive oxygen/nitrogen species (RONS) is based on enzyme-catalyzed reactions, efficient recycling pathways, as well as the mass effect achieved by its high intracellular concentrations (2–10 mM). Among the important enzyme-mediated cellular processes that use GSH as a cofactor is the GSH-dependent reduction of hydrogen peroxide and other peroxides catalyzed by glutathione peroxidase (Figure 2). Hydrogen peroxide can also be destroyed by catalase through a dismutation reaction [4]. Cells, especially erythrocytes, are extremely rich in catalase (>300,000 U/mg protein), but the role of GSH peroxidase under physiological conditions is not insignificant, as shown by experiments with fibroblasts, which demonstrated that 80% to 90% of H_2_O_2_ is degraded by GSH peroxidase at H_2_O_2_ concentrations below 10 μM [5]. GSH forms *S*-conjugates with either xenobiotics (or their metabolites) or endogenous compounds. These reactions are usually catalyzed by a family of isoenzymes called glutathione *S*-transferases [6] (for a review). However, there are also some examples where the reaction occurs spontaneously. In the reaction with free radicals, GSH generates a sulfur-centered thiyl radical (reaction 1):GSH + R^●^ → GS^●^ + RH(1)
2GS^●^ → GSSG(2)
GS^●^ + GSH → GSSG^●^
(3)
GSSG^●^ + O_2_ → O_2_^●− +^GSSG (4)

The thiyl radical is rather stable and can usually react in the presence of oxygen either with another thiyl radical or with another GSH molecule to form a disulfide radical. This, in turn, generates a superoxide anion in the presence of oxygen (reactions 2–4). In both cases, glutathione disulfide (GSSG) is eventually formed [7].

The availability of GSH is ensured by its recycling and biosynthesis, which can be upregulated in situations of depletion due to oxidative/nitrosative stress or *S*-conjugation with electrophiles. In summary, intracellular GSH levels are influenced by several factors: (i) oxidative stress, (ii) conjugation with electrophiles, (iii) export from the cell, (iv) enzymatic reduction of GSSG, and (v) de novo synthesis. Reactions i–iii can be classified as depleting GSH reactions, whereas iv–v are considered as replenishing GSH reactions. The decrease in GSH may be due to reduced GSH production because of certain pathophysiological conditions as well as drug treatment. In addition, GSH levels may also be determined by nutritional status, some hormones, psychogenic stress, and physical activity and may also exhibit diurnal variations [8]. Evidently, under oxidative stress, a significant percentage of GSH is converted to GSSG, and this also represents a situation in which GSH is depleted. In mammalian cells, glutathione is present mainly (>99%) in the reduced form (GSH), whereas GSSG normally constitutes < 1% of GSH. Hundreds of papers report GSSG concentrations much higher than 1% of GSH, likely due to artifacts in sample manipulation (i.e., oxidation) [9,10,11]. This is mainly due to acid precipitation of proteins, which is crucial for GSH determination. Under these conditions, a significant percentage of GSH tends to oxidize. This strongly affects the measured GSH values, which can be 1–2 orders of magnitude higher compared to the actual ones. This phenomenon occurs primarily in blood, where hemoglobin plays a central role in the formation of this artifact [12] (for a review).

A minimal percentage of GSH can also be found to be protein-bound (GSSP); this percentage, in parallel with GSSG, can be increased by pro-oxidant conditions. The formation of GSSP may play an important role in the allosteric regulation of protein function. This process, called protein *S*-glutathionylation, results in the inhibition of enzymes carrying—SH in the catalytic center, allosteric modifications by introducing a negative charge into the protein (Glu residue), and protection of protein—SH groups from further oxidation [13]. As the *S*-glutathionylation of proteins is mainly due to transglutathionylation reactions (5), whose equilibria are close to unity, and the cellular GSH/GSSG ratio seems to play a central role:PSH + GSSG ↔ GSSP + GSH (5)

Although these reactions may occur spontaneously, the nonenzymatic reaction in vivo appears to be negligible [14]. In general, cellular thiols other than GSH, such as cysteine (Cys), coenzyme A, and, more importantly, critical protein thiols, are maintained in a reduced state by this mechanism involving the two enzymes thioltransferase and glutaredoxin. Therefore, reaction (5) can be rewritten as follows:RSH + GSSG ↔ RSSG + GSH (6)
where RSH can be either a low molecular weight thiol or a protein thiol. Thioltransferase has been reported to have broad specificity ranging from asymmetric low molecular weight disulfides to mixed protein disulfides. It is unlikely that only those protein disulfide groups that are sterically inaccessible will react. It should be emphasized that thioltransferase can maintain several enzymes such as phosphofructokinase and pyruvate kinase in the reduced active state [15].

Normally, the GSH/GSSG ratio is finely regulated in the GSSG formed after the reaction of GSH with oxidants can be reduced back to GSH by glutathione reductase (GR) with NADPH donating reducing equivalents. Riboflavin is also required in this reaction because flavin dinucleotide (FAD) forms the prosthetic group of GR [16]; therefore, its deficiency may lead to an increase in GSSG. 

Increasing the GSH/GSSG ratio appears to be critical for cell survival and for the regulation of the cell cycle [17], which is why the de novo synthesis of GSH, reduction of GSSG to GSH, and export of GSSG are considered protective [18].

GSH competes with other cellular redox-active biomolecules for the reduction of reactive oxygen species and thereby exerts its well-known protective function. However, GSH is also involved in several vital functions in animals: (i) it is essential in some phase IIs of xenobiotic metabolism; (ii) it may provide a reservoir for nitric-oxide-forming S-nitrosoglutathione [19], which is essential for the regulation of blood pressure; (iii) it is necessary for the conversion of prostaglandin H2 (a metabolite of arachidonic acid) into prostaglandins D2 and for leukotriene synthesis; (iv) there is evidence both in vitro and in vivo that GSH inhibits influenza virus infection [20]; (v) SARS-CoV2 infection impairs cellular glutathione metabolism and redox function, thus the replenishment of GSH during infection is thought to play a protective role against disease [21].

Last but not least, a decrease in GSH/GSSG activates several signaling pathways that reduce cell proliferation and induce apoptosis [22]. It follows that a decrease in GSH can impair normal physiological functions and cell survival [18] (for a review).

## 3. Synthesis and Catabolism of Glutathione

GSH is synthesized intracellularly by the sequential actions of γ-glutamylcysteine and GSH synthetases (γ-GCS and GSHS, respectively). The first step is of central importance: the activity of γ-GCS is subject to nonallosteric inhibition by GSH (Ki = 2.1 mM), and the availability of cysteine drives the kinetics of the whole reaction (Figure 3). Cysteine is mainly derived from the trans-sulfuration pathway (TSP) of methionine and/or the reduction of cysteine [23]. 

GSH is degraded extracellularly, where it is converted to cysteinylglycine (CysGly) and then to Cys at tissue sites rich in the ectoenzymes γ-glutamyltranspeptidase (γ-GT) and dipeptidases (mainly kidney and lung) by the sequential action of these two enzymes Figure 3) [24] These amino acids can be uploaded by cells and used for de novo GSH biosynthesis. GSH turnover is relatively rapid in most cells, with half-lives as short as 2–6 h [25], indicating high rates of both GSH synthesis and export. In addition to GSH, GSSG and GSH conjugates can also be exported when formed by cellular exposure to oxidants or electrophilic molecules. It is known that the export of glutathione conjugates from cells occurs primarily via proteins associated with multidrug resistance. Less is known about the export of GSH and GSSG, but some results suggest that the phenomenon is mediated by the same class of transporter proteins [26]. GSH appears to be released from all tissues, particularly the liver (and to a lesser extent skeletal muscle and red blood cells), where cysteine derived from food or methionine via TSP is converted to tripeptide.

## 4. GSH Levels in Tissues 

GSH is present in all organs, with the highest concentration found in the liver, where it ranges from 6 to 8 mM in both rats (Table 1) and mice [27]. However, GSH levels are also above 1 mM in all other organs (Table 1). Instead, low micromolar GSH concentrations occur in adipose tissue [28]. In cultured cells, GSH levels vary between 20 and 150 nmol/mg protein [29,30].

Unlike in the intracellular compartment, GSH is present in the extracellular fluids at low micromolar concentrations (Table 1), and, in humans, it is almost equimolar with GSSG [31]. In contrast to humans, rat plasma has higher levels of GSH, i.e., about 20 μM, which far exceeds GSSG (GSH/GSSG ratio = 6) [32]. Glutathione is present in very low amounts in the extracellular cortex, whereas its concentration is much higher in the epithelial mucosal fluid of healthy humans, where it is important in maintaining the fluidity of the mucus by reducing protein disulfide bridges [33]. 

**Table 1 antioxidants-12-01094-t001:** Levels of GSH measured in different tissues and cells.

Tissue	Species	Concentration	Reference
Kidney	Human	1564 ± 106 μM	[34]
Muscle	Human	2543 ± 267 μM	[35]
RBCs	Human	8470 ± 1750 nmol/g Hb	[36]
Neutrophils	Humans	13.2 ± 1.8 nmol/10^7^ cells	[37]
Lymphocytes	Human	5.7 ± 0.35 nmol/10^7^ cells	[38]
Platelets	Human	13.4 ± 2.64 nmol/10^9^ platelets	[12]
Plasma	Human	3.1 ± 0.26 μM	[31]
ELF	Human	0.2–0.4 mM	[33]
Liver	Rat	8221 ± 558 μM	[39]
Kidney	Rat	2221 ± 302 μM	[39]
Lung	Rat	2314 ± 182 μM	[39]
Heart	Rat	1835 ± 244 μM	[39]
Spleen	Rat	2648 ± 55 μM	[39]
Brain	Rat	1801 ± 59 μM	[39]
Extracellular brain cortex	Rat	2.10 ± 1.78 μM	[40]
White adipose tissue	Mouse	~4.3 nmol/mg protein	[28]
Brown adipose tissue	Mouse	~2 nmol/mg protein	[28]

Abbreviations: RBCs, red blood cells, ELF, epithelial lining fluid, Hb, hemoglobin.

## 5. GSH and Disease

GSH plays a central role in cellular functions and viability, and its levels are very tightly regulated. Therefore, a significant decrease in its concentration can cause severe pathological manifestations. This is especially true for people suffering from genetic diseases caused by congenital defects in GSH metabolism. The relationship between oxidative stress, GSH depletion, and disease is much less clear. In this case, the decrease in GSH levels is due to its oxidation caused by the overproduction of RONS. Due to the physiological regulation of homeostatic GSH concentration, it is not easy to identify those pathological conditions in which oxidative stress may have this effect. In any case, much research has been performed on this topic, and some data suggest that this relationship does exist.

### 5.1. Inborn Alterations in GSH Metabolism 

GSH levels are fine-tuned by the activity of several enzymes involved in both its synthesis and metabolism. Although these are rare events, there is evidence that deficiency of any of these enzymes can lead to a significant drop in GSH levels that is incompatible with life in most cases [41]. Thus, the deficiency of γ-GCS, the enzyme that catalyzes the first step of GSH synthesis, is a rare autosomal recessive disease characterized by hemolytic anemia and, in a smaller number of cases, neurological disorders. Similarly, deficiency of glutathione synthetase, which catalyzes the addition of glycine to glutamylcysteine to form GSH, also causes hemolytic anemia. Patients with more severe deficiency experience recurrent bacterial infections and neurodegenerative diseases. In both cases, GSH levels are far below physiological levels. Low activity of GSSG reductase results in low GSH levels and hemolytic anemia [41]. 

### 5.2. Ageing and Related Diseases 

The free radical theory suggests that age-related loss of function is the result of cumulative damage by endogenous radicals released primarily by mitochondria. RONS trigger various effects, such as the induction of matrix metalloproteases, pro-inflammatory interleukins, and the activation of some specific signaling pathways responsible for organ-related senescence [42]. In particular, a close relationship has been suggested between aging, oxidative stress, and inflammation, a vicious cycle in which chronic oxidative stress and inflammation reinforce each other and, consequently, increase age-related morbidity. Age-related diseases such as diabetes, cardiovascular disease, chronic obstructive pulmonary disease (COPD), cognitive function loss, macular degeneration, and sarcopenia are thought to have oxidative stress as a cause. Considering the protective properties of GSH against oxidative damage, the deficiency of GSH may be responsible for the onset and/or progression of these diseases.

#### 5.2.1. Age-Related Ocular Diseases

A local decrease in GSH has been noted in age-related eye diseases such as cataract, where a direct correlation between GSH loss and lens browning has been observed [43]. Indeed, lens transparency requires that the thiol group of the structural proteins is in the reduced redox form, and the formation of high molecular weight agglomerates is attributed to the overproduction of disulfide bonds. Therefore, GSH is thought to be necessary to maintain the cysteine residues of proteins in the reduced form. GSH also plays an important homeostatic role in the retina as it is exposed to sunlight, leading to an increase in RONS formation. For example, a decrease in GSH was measured in the retinal pigment epithelium of mice after exposure to white fluorescent light (8000 lux) for 2 h. This observation has been linked to the development and progression of age-related macular degeneration in humans [44].

#### 5.2.2. COPD 

COPD is characterized by chronic progressive airway obstruction, with local inflammation playing an important role and tobacco smoke acting as a cofactor. Collectively, these etiologic factors may cause local oxidative injury and significant GSH depletion. Indeed, GSH content in alveolar cells has been shown in animal studies to be decreased after acute exposure to cigarette smoke [45]. GSH concentration is also decreased in COPD patients [46], which indirectly correlates with the severity of respiratory disease [47].

#### 5.2.3. Diabetes Mellitus 

Oxidative stress is involved in the pathogenesis of diabetes mellitus and its micro (e.g., retinopathy, neuropathy, and nephropathy) and macrovascular (essentially cardiovascular) complications. Several mechanisms may be responsible for this, such as the autoxidation of glucose, the increased formation of advanced glycation and lipoxidation end products, and the increase in polyol pathway flux, which, in turn, consumes NADPH [48]. As an additional mechanism, there is also some evidence that the activity of antioxidant enzymes can be modulated by glycation. This information has been obtained, for example, from in vitro experiments with lens cells exposed to glucose [49]. As hyperglycemia occurs in both type I and type II diabetes, oxidative stress is considered a pathogenic factor in both cases. Indeed, there are several clinical studies showing a significant decrease in blood GSH levels in type I and II diabetics [50,51]. Our research group has also found a significant increase in oxidized forms of GSH in the erythrocytes of patients with type I diabetes [52].

#### 5.2.4. Cardiovascular Diseases 

Inflammation plays an important role in the pathogenesis of atherosclerosis, which is triggered by endothelial injury, which, in turn, promotes compensatory responses and, in particular, the recruitment of inflammatory cells in the arterial wall. Therefore, atherosclerosis is characterized by a local increase in RONS and a decrease in antioxidants. Several studies, either in animal models or in humans, support the notion that the decrease in GSH can lead to cardiovascular events such as stroke [53] and cardiac transplantation [54]. Interestingly, Julius et al. demonstrated that elderly people affected by arthritis, diabetes, or heart disease had lower blood GSH levels than healthy age-matched individuals [55].

#### 5.2.5. Neurodegenerative Diseases 

GSH is central to brain function, as this organ consumes a high percentage of oxygen to produce ATP. Some RONS are formed during this mitochondrial process but are counteracted by intracellular antioxidants. Several neurodegenerative diseases are characterized by mitochondrial degeneration associated with an increase in ROS production and oxidative damage. Evidence that GSH plays a neuroprotective role was confirmed by data from Sofic et al., who measured it in the substantia nigra of people with Parkinson’s disease (PD) [56]. The fact that oxidative stress plays an important role in PD was also confirmed by the evidence that paraquat, a free radical generating herbicide, is able to induce the degeneration of dopaminergic neurons and PD in mice [57,58]. It has been reported that antioxidant defenses in the substantia nigra are relatively low compared with other regions of the central nervous system, due to low GSH levels, especially in the early stages of PD, when extravescicular dopamine and its degradation products may act as GSH-depleting agents. These neurons are protected by restoring normal GSH levels [59]. 

#### 5.2.6. Other Age-Related Conditions

Unlike in younger people, a slight reduction in protein intake (less than 10% of calories) has been observed to increase the risk of death in people over 65 years of age [60]. No explanation for this observation has yet been found, but, among the various hypotheses, the possibility that some specific amino acids are essential for health, and, in particular, cysteine, the precursor of GSH, has also been considered. As GSH decreases with age, maintaining physiological GSH levels may protect against general health worsening due to tissue deterioration (e.g., muscle atrophy, osteoporosis, etc.).

### 5.3. Cystic Fibrosis 

Cystic fibrosis is a genetic disease characterized by the chronic inflammation of the lungs leading to progressive damage to the airway epithelium, bronchiectasis, and chronic obstructive pulmonary disease. The etiology of the disease is due to a mutation of the transmembrane chloride channel CFTR, which is mainly expressed in the apical membrane of the epithelial cells of the lung. However, it is generally accepted that disease progression is favored by the local release of RONS from inflammatory cells. GSH is found in the epithelial lining fluid of the airways, where it protects the lungs from oxidative damage. There are several data indicating a decrease in GSH levels in lung epithelial fluid, plasma, and neutrophil granulocytes in the blood of patients with cystic fibrosis [61,62,63]. In addition, defective CFTR in lung epithelial cells has been shown to affect the transport of GSH within the cells, contributing to its decrease [64]. 

### 5.4. Other Diseases

The above are just a few examples of pathological conditions in which it is more likely that GSH depletion may play a role either as a primary cause or as a secondary contributor to disease progression. Indeed, oxidative stress and GSH decrease are associated with a wide variety of diseases. In recent decades, a wealth of research has emerged, demonstrating a link between reduced GSH concentration and the pathogenesis or progression of almost all known human pathological conditions [65] (for a review).

## 6. GSH Levels Regulation 

There is mounting interest in identifying the best method(s) to increase cellular glutathione for both disease prevention and treatment. GSH concentrations in cells appear to be very finely tuned. Briefly, the substrates for de novo synthesis, impairment of GSH export, and activation of Nuclear Factor Erythroid 2-Related Factor 2 (Nrf2) leading to induction of γ-GCS and GSHS could be the main targets of treatments if the goal is to increase GSH concentration in cells (Table 2).

### 6.1. GSH, GSH Esters, γ-Glutamylcysteine

The direct administration of GSH is probably the most evident method to increase GSH. However, orally administered GSH has low bioavailability because it is broken down in the gut into the amino acids of which it is composed. In addition, GSH is not taken up by cells to any significant extent. To circumvent these problems, Shina et al. investigated the efficacy of oral supplementation with liposomal GSH in humans. Although statistical power was limited due to the small sample size, this approach appeared promising [66]. Other routes of GSH administration, such as intravenous, intranasal, and novel sublingual formulations, have been investigated. However, these treatments appeared to be problematic or ineffective as GSH enhancers for a variety of reasons [67].

To improve bioavailability and cell entry, esters of GSH have been synthesized and studied. After the intraperitoneal administration of GSH ethyl ester (GSH-EE), an increase in GSH content was observed in various tissues, including cerebrospinal fluid and erythrocytes. The oral administration of the ester to mice also increased cellular GSH content in their kidneys and livers after it had been decreased by treatment of the animals with buthionine sulfoximine (an inhibitor of GSH synthesis) [73,83]. However, in experiments with endothelial cells (HUVEC), we did not find such efficacy in increasing GSH levels [84]. It was also found that GSH diesters are effectively transported into cells and cause an increase in intracellular GSH concentrations [85]. However, even if diesters are rapidly transported into and out of cells, they are rapidly cleaved into GSH monoesters, which are transported more slowly than GSH diesters. Preliminary studies by Levy et al. suggested that the GSH diester was about four times more effective than the GSH monoester in raising GSH levels in hamster livers [85].

γ-Glutamylcysteine (γ-Glu-Cys) is an immediate precursor of GSH, and only the catalysis of glutathione synthetase is required for the formation of glutathione. In a recent clinical study, γ-Glu-Cys was shown to successfully increase GSH levels in lymphocytes of healthy humans when administered orally (2 g) [68]. The results were promising, although the number of subjects participating in the study was small (thirteen); thus more data are needed to confirm this observation. It was also shown that exogenous γ-Glu-Cys could easily cross the blood–brain barrier (BBB) and be taken up by many cell types. Liu et al. found that dietary supplementation with γ-Glu-Cys improved spatial memory in mice with Alzheimer’s disease by increasing total GSH and reducing levels of oxidative stress and nerve cell apoptosis [74].

### 6.2. Nrf2 

Nrf2 is a redox-sensitive transcription factor that binds to specific response elements in promoter sequences and drives the expression of numerous cytoprotective genes involved in the antioxidant response. Enzymes upregulated by Nrf2 include superoxide dismutase, enzymes involved in glutathione production (γ-GCS, GSHS), glutathione peroxidase, thioredoxin, the X_c_¯ system that facilitates cystine uptake, and many others with direct or indirect antioxidant function [86,87]. Nrf2 resides in the cytoplasm under basal conditions and has a short half-life, as it is ubiquinated by Kelch like-ECH-associated protein 1 (KEAP1) and cullin 3 [88]. After ubiquination, it is rapidly destroyed. Electrophilic molecules or oxidative stress oxidize the cysteinyl residues in Keap1, inhibiting the Keap1-Cul3 ubiquitination system [89]. Therefore, molecules that can activate Nrf2 have been the focus of interest due to their potential as GSH-enhancing drugs. Resveratrol, sulforaphane, lipoic acid, bardoxolone, curcumin, and dimethylfumaric acid have been shown to be Nrf2 activators [90,91,92,93]. The molecular mechanisms of action of these molecules, most of which are antioxidants (e.g., resveratrol) rather than oxidants, are unclear. It has been hypothesized that they may cause the indirect oxidation of a specific thiol residue in Keap1, leading to the stabilization of Nrf2 by inhibiting its proteasomal degradation and subsequent translocation to the nucleus [91,94,95]. Triterpendoids such as bardoxolone have been reported to induce cytoprotective genes through Keap1-Nrf2 antioxidant response element signaling and, in particular, increase cellular GSH levels [82]. Bardoxolone has been tested for the treatment of diseases in which oxidative stress is thought to play a key role, such as diabetic nephropathy. However, a phase 3 clinical trial evaluating bardoxolone for the treatment of chronic kidney disease was discontinued after patients were found to have a higher rate of heart-related adverse events [96]. Currently, the efficacy and safety of bardoxolone is being evaluated in the treatment of pulmonary hypertension [97].

Dimethyl fumarate has been approved by the US Food and Drug Administration as a treatment option for adults with relapsing–remitting multiple sclerosis [98]. Oral treatments with dimethyl fumarate appear to be able to increase GSH levels through the activation of Nrf2, leading to protection against oxidative stress [77]. As oxidative stress plays an important role in neurodegeneration in multiple sclerosis, increasing GSH levels seems to be a reasonable therapeutic option [99]. In 2015, a Cochrane systematic review found moderate-quality evidence of a reduction in the number of people with relapsing-remitting MS, who relapsed over a 2-year treatment period on dimethyl fumarate compared with placebo; low-quality evidence of a reduction in disability worsening; and a general need for higher-quality trials with longer follow-up [100].

### 6.3. Cysteine Pro-Drugs

As mentioned earlier, cysteine availability is a rate-limiting factor for GSH synthesis. Therefore, treatments that can increase cellular cysteine levels are considered an appropriate strategy to increase GSH. To this end, several cysteine prodrugs have been developed that are more suitable than cysteine itself, given its high chemical instability and tendency to oxidize to the highly insoluble cystine. 

#### 6.3.1. N-Acetylcysteine (NAC) 

The best-studied cysteine pro-drug is NAC, which has been used in clinical practice for several decades both as an expectorant and as emergency therapy to treat acetaminophen intoxication. In addition to these specific applications, NAC is also being investigated as a potential therapeutic agent for all diseases in which oxidative stress and the resulting decrease in GSH play a role. The efficacy of NAC in increasing intracellular GSH levels is due to its de-acetylation to cysteine within cells by various aminoacylases. The kinetics of this process were investigated in an in vitro study [101]. The results suggest that oral administration likely provides little cysteine for accelerated GSH synthesis, due to both poor uptake and minimal intracellular de-acetylation. However, an indirect mechanism could be the reaction of NAC with extracellular cysteine-containing disulfides (i.e., cystine, mixed disulfides with proteins, mixed disulfides with other low molecular weight thiols), allowing it to be released and enter cells [84]. NAC appears to function well as a GSH precursor in animal studies and in small clinical trials for a variety of indications, but, often, when studies were repeated, results were inconsistent [102] (for review). One reason for this could be its low bioavailability. In fact, it is about 10% after oral ingestion [80,103]. Numerous in vitro experiments (e.g., with bovine pulmonary artery endothelial cells, Chinese hamster ovary cells [104,105]) and in vivo data indicate the efficacy of NAC in increasing GSH in cells. An 8-week oral treatment with NAC (8 g/day in divided doses) was effective in restoring GSH levels in the whole blood and T cells of HIV-infected patients [69]. However, in another clinical trial, no change was observed in peripheral blood mononuclear cells (PBMC) of the healthy control group and only a moderate increase in the HIV-positive group [106]. The effect of the intravenous administration of NAC at various doses was compared between patients with known idiopathic pulmonary fibrosis and healthy subjects, and no difference in GSH levels was found in the healthy subjects group [107], supporting the concept that NAC has an effect only on replenishing GSH levels in tissues deficient in the tripeptide. Higher GSH levels were also found in the blood of autism patients after oral treatment with NAC (60 mg/kg/day in divided doses, 12 weeks). A similar effect was observed in the blood of patients with cystic fibrosis [63] and diabetes [108]. There are two aspects to consider when using NAC to replenish intracellular GSH levels: (a) Is NAC equally effective in increasing GSH levels in the specific organs damaged by a pathological condition? (b) Do the typical pharmacokinetics of NAC possibly require too high of a dosage to achieve a significant effect on GSH that patients are unlikely to take for an extended period of time? In experiments conducted in our laboratory with rats, 50 mg/kg NAC (*per os*) twice daily for a period of two weeks was ineffective for increasing GSH levels in most organs analyzed (including the brain) [80], but this was in contrast to other results [109]. Our data also shed light on another aspect that is not well studied: Is NAC able to increase GSH content in the central nervous system because the blood–brain barrier hinders the delivery of NAC to the central nervous system? 

Supplementation of NAC and glycine (GlyNAC) has been proposed as an alternative strategy to NAC administration [110,111]. This idea stems from the observation that GSH concentration in the blood of the elderly is low compared to younger control subjects. The same authors found that two of the three amino acids that make up GSH, glycine, and cysteine were significantly lower compared to younger subjects and that this could be responsible for GSH deficiency. The simultaneous administration of glycine and NAC at a fixed dose (1.33 mol/kg/day and 0.81 mmol/kg/day, respectively) over a 24-week period to elderly subjects was able to correct GSH depletion in red blood cells and reduce some hematological markers of oxidative stress such as F2-isoprostanes. A summary of the observed effects of this supplement on age-related defects and the profile of tolerability and safety is provided in a recent report by Sekhar [112].

#### 6.3.2. N-Acetylcysteine Ethyl Ester (NACET) 

As an alternative to NAC and with the aim of improving its pharmacokinetic properties, we developed and studied NACET. This molecule was found to cross the plasma membrane much better than NAC with a higher hydrophobicity (logD = −5.4 for NAC and 0.85 for NACET) [80]. NACET was also characterized by a 10-fold bioavailability compared to NAC. Taken orally, it is rapidly absorbed by the GI tract and enters cells as a lipophilic substance. In the cell, it is de-esterified to the more hydrophilic NAC, which, in turn, is trapped in the cell and slowly converted to cysteine, acting as a cysteine pro-drug. After IV bolus administration to rats, the greatest amount of NACET was found in the cells of various tissues, including the brain, mainly as NAC and Cys. Similarly, the chronic treatment of rats with equivalent doses of NAC or NACET showed that only NACET was able to significantly increase the GSH content of most tissues studied, including the brain [80]. Due to these properties, NACET is much more effective than NAC as a GSH enhancer. Comparative in vitro experiments in which HUVEC were treated with pharmacological concentrations of different GSH boosters (NAC, GSH monoethyl ester, oxothiazolidine-4-carboxylic acid) showed that only NACET was able to significantly influence intracellular GSH [84]. A similar effect was obtained with other cells, i.e., cells of the retinal pigment epithelium (ARPE-19 cells). In the same study, rats were orally administered 50 mg/kg NAC or NACET, and the GSH concentration in the eyes was measured. GSH levels were significantly increased by NACET treatment, with a peak 4 h after drug administration. In contrast to NACET, NAC had no significant effect on GSH concentration [75]. These data indicate that NACET is a promising alternative to other drugs for increasing intracellular GSH concentration, thanks to its best pharmacokinetic properties. It is desirable that NACET continues its preclinical characterization and hopefully can be tested in clinical trials in the future to confirm these positive properties.

#### 6.3.3. N-Acetylcysteinamide (NACA) 

An amide derivative of NAC, namely, N-acetylcysteinamide (NACA), was also developed by modifying the carboxyl group. This is another way to improve the lipophilicity of NAC and thus optimize its pharmacokinetics. NACA showed high bioavailability (about 60%), but it was assumed that it was metabolized to NAC in a first step [113]. NACA has been shown to be able to restore basal levels of GSH in erythrocytes or cultured cells treated with oxidants [114].

#### 6.3.4. Thiazolidines 

Another class of compounds that can increase intracellular cysteine is thiazolidines. The first observation of these molecules comes from the L-thiazolidine-4-carboxylic acid, which can be converted to N-formyl-cysteine by proline oxidase. Cysteine can then be generated by hydrolysis. The best studied molecule of this class is 2-oxothiazolidine-4-carboxylate (OTC), which, when tested in isolated hepatocytes, showed a low uptake rate and slow conversion to Cys once it entered the cells [115]. However, when administered by IV infusion to patients with acute respiratory distress syndrome (63 mg/kg every 8 h for 10 days), the replenishing effect on GSH in red blood cells was similar to that of NAC [116]. OTC (500 mg orally, every 8 h for 14 days) increased, circulating GSH levels in patients on peritoneal dialysis [70]. However, as indicated above, in HUVEC cells, treatment with OTC for 12–72 h did not significantly increase intracellular GSH [84].

#### 6.3.5. Other Cysteine Derivatives

S-ethylcysteine, S-methylcysteine, and S-propylcysteine increased GSH levels in the striatum and reduced MPTP-induced oxidative stress, inflammatory damage, and loss of dopaminergic neurons in an animal model of Parkinson’s disease. These compounds added to drinking water (1 g/L for 3 weeks) showed antioxidant and anti-inflammatory activities by acting as GSH precursors [76]. S-allylcysteine, as well as S-ethylcysteine, S-methylcysteine, and S-propylcysteine, increased GSH content in the kidneys and livers of orally treated Balb/c mice. In addition, these compounds had significant effects on antioxidant enzymes and spared alpha-tocopherol [117].

### 6.4. Taurine

Recently, taurine was shown to provide protection against oxidative stress by increasing glutathione stores [118,119]. Similarly, high-dose taurine treatment (400 mg/kg via oral gavage for 9 weeks) was able to increase GSH content in whole-brain homogenate of rats exposed to D-galactose [78]. The effects of taurine on GSH levels can be explained by the fact that cysteine is a precursor of both molecules (GSH and taurine); thus, sparing cellular cysteine in the provision of taurine may lead to increased production of GSH. In this context, it is helpful to remember that the availability of cysteine determines the rate of cellular GSH production.

### 6.5. Silymarin 

Silymarin is a mixture of flavonolignans extracted from the milk thistle *Silybum marianum*. It has been shown to prevent liver damage caused by various chemicals or toxins. This natural herbal medicine has been successfully used for centuries to treat liver diseases [120]. Silymarin acts as an antioxidant but also has a number of other biological properties, such as reducing inflammatory responses by inhibiting NF-κB signaling pathways [121] (for a review). The hepatoprotective role of silymarin against xenobiotic electrolytes such as carbon tetrachloride, ethanol, diethylnitrosamine, and cisplatin is well established [122,123]. Silymarin is a direct scavenger of many RONS, but it is also able to increase the levels of several antioxidant enzymes and of GSH [79]. It has also been shown to significantly increase Nrf2 protein levels in CON1 cells after 48 h of treatment [124]. The main mechanisms by which silymarin increases GSH levels are both the upregulation of Nrf2 and the increase in cysteine availability by increasing its synthesis and inhibiting its degradation to taurine [125]. In summary, silymarin spares the direct reaction of GSH with RONS, upregulates Nrf2, and increases cysteine levels; all of these actions combine to increase GSH levels in cells.

### 6.6. Food and Diet

There is growing evidence that diet may be an alternative to drugs to increase GSH levels. First of all, given the chemical structure of GSH, it is possible that its levels can be increased by ingesting dietary proteins that contain its precursors, particularly cysteine. In this context, bovine whey, a byproduct of cheese production that is particularly rich in cystine-containing proteins, is thought to play an important role. Moreover, bovine whey is not only rich in cysteine-containing proteins [126], but also contains significant amounts of the GSH precursor **γ**-glutamilcysteine. It can thus provide a good source of cysteine without the disadvantages of direct supplementation of cysteine or cystine, i.e., low solubility and toxicity. To date, several clinical studies have been conducted confirming the GSH-enhancing effect of whey protein isolates taken over several weeks by healthy individuals [127] or by individuals with various pathologies (e.g., nonalcoholic steatohepatitis, cancer) [128,129]. A pilot clinical study with Parkinson’s’ disease patients treated with a whey protein supplement showed that the treatment significantly increased plasma GSH in Parkinson’s patients but not in the control group [130]. 

Immunocal^®^ is an undenatured whey protein supplement designed to augment the available cellular GSH pool. Immunocal^®^ has been shown to reduce oxidative stress by increasing lymphocyte GSH in both HIV-infected patients and cystic fibrosis patients [131].

Various nutrients (such as vitamins and minerals) have also been studied for their ability to increase GSH levels. For example, vitamin B6 could increase GSH levels because it is a cofactor for enzymes involved in the TSP, the metabolic pathway that converts homocysteine to cysteine. Vitamins C and E, as well as selenium, have also been tested for their antioxidant effects, which may be mediated by increasing GSH levels, but there is no clear clinical evidence to date [132]. Daily treatment with 500 mg of L-ascorbate for two weeks increased mean glutathione in red blood cells by nearly 50% compared to baseline in healthy, nonsmoking volunteers [133]. Ascorbic acid supplementation in a group of healthy ascorbic acid-deficient volunteers was associated with an 18% increase in lymphocyte GSH [134]. These experiments suggested that each change of 1 mole of ascorbate was accompanied by a change of about 0.5 mole of GSH. Presumably, ascorbate conserved glutathione by limiting its reaction with RONS. The lack of efficient biochemical systems to regenerate GSH when it was oxidized to species other than disulfides (e.g., sulfonic acids) may have led to its depletion. Ascorbic acid competes with GSH by acting as an electron donor, but it also forms semidehydroascorbyl radicals, which are relatively unreactive and are efficiently reduced to ascorbate by specific NADPH-dependent reductases [135]. This may be the reason why high concentrations of vitamin C enhance GSH.

Some foods may contribute to the maintenance of physiological GSH levels. First, some fruits and vegetables contain significant amounts of GSH (e.g., asparagus, avocado, spinach) or its precursor cysteine (asparagus, red bell pepper, papaya) [136]. However, it is also clear from this study that there is a large heterogeneity between different food types and that the contribution of other aspects such as storage, preparation, and seasonal differences has not yet been elucidated. The effect of cooking was evaluated in a previous article and showed that this process could reduce GSH content. In the same study, it was also found that GSH may be present in foods as glutathione disulfide, which could be reduced back to GSH in cells by GR [137]. Therefore, this point is also worth considering.

Cruciferous vegetables are also of interest because they contain molecules capable of increasing GSH, such as sulforaphane and dithiolethiones. Since the 1980s, it has been observed that some dithiolthione compounds, such as anethole dithiolthione (ADT) and oltipraz, were able to increase GSH content in various organs of rats and mice and protect against the toxic effects of acetaminophen [138]. More recently, we have confirmed this GSH-replenishing effect, especially in different organs of rats such as the liver, kidneys and brain [81]. The molecular mechanisms of this effect remain unclear. We have observed a decrease in the activity of γ-GT in rat kidneys, which correlated with an increase in GSH in the same tissue, especially in the cells of the proximal tubule [139]. It could be speculated that the inhibition of γ-GT by ADT leads to a decreased degradation of GSH and induces the reuptake of intact GSH by the proximal tubule cells. There is also increasing evidence that ADT is able to stimulate the Nrf2 signaling pathway, but this point has not been fully elucidated [140]. The GSH-enhancing effect for this class of compounds was also demonstrated in a clinical study with oltipraz, which was able to increase lymphocyte glutathione in healthy humans [141]. To date, however, there is no direct evidence as to whether a cruciferous vegetable diet could affect physiological GSH levels. Other foods are also of interest because of their antioxidant properties, e.g., polyphenol-containing foods such as green tea and fruit juices, but there are few data on their effects on GSH levels [71].

Interestingly, there is some evidence that the adherence to specific dietary approaches such as the DASH diet (Dietary Approaches to Stop Hypertension) can successfully increase blood GSH levels. This was recently demonstrated in a meta-analysis that examined the effects of this type of diet on some oxidative stress biomarkers [72]. Among the parameters analyzed, GSH was one of those increased in the patients who followed this diet, which consisted mainly of fruits, vegetables, low-fat dairy products, whole grains, fish, poultry, and nuts [142]. However, considering that the participants of the clinical trials selected in this meta-analysis suffered from various diseases such as hypertension, obesity, and polycystic ovary syndrome, and the number of subjects studied was not very high (8 trials, 217 patients), further well-designed studies are needed to confirm this potential beneficial effect.

## 7. Conclusions and Future Perspectives

Undoubtedly, there is a great and growing interest in molecules that increase GSH levels due to their antioxidant and detoxifying effects. Nevertheless, some of the mechanisms involved remain enigmatic, and, more importantly, for many molecules, the available data are inconsistent. The most commonly used molecule NAC appears to perform this function in some experiments but fails in others or is effective only at very high doses. Other approaches using cysteine prodrugs similar to NAC are also promising but require further research. In particular, we have found that NACET is able to rapidly increase GSH in most tissues and cultured cells because of its unique pharmacokinetic properties (it is deesterified to NAC and deacetylated to Cys). Although NACET is an excellent cysteine precursor, we observed that high concentrations of NAC competitively inhibited γ-GCS, so that it may paradoxically exert a hormetic effect once converted to NAC [84]. The same aspect should be considered for other NAC donors.

Molecules that act via Nrf2 activation and, in turn, affect the expression of various antioxidant enzymes, including those required for GSH production, have been extensively studied in detail. However, the mechanism of Nrf2 activation by these molecules remains largely unexplained. These issues contribute to the high degree of uncertainty surrounding the best conditions for treating animals and/or humans (i.e., dose, duration of treatment, frequency of administration in long-term studies) to achieve the desired effect. Finally, it should be remembered that cells tightly regulate GSH levels, and, even if treatment causes a transient increase in GSH concentration, both the allosteric inhibition of γ-GCS by GSH and the increased GSH export cause cellular GSH levels to return to pretreatment levels. In this sense, little is known about the importance and regulation of GSH exports. This is another aspect that may account for the heterogeneity of the data obtained thus far. Indeed, the experimental results obtained may change depending on whether the treatments are performed with cells/animals having either normal or depleted GSH levels or with healthy or diseased humans with probably different GSH concentrations.

In addition, as a general concept, it also needs to be clarified whether there is sufficient evidence that increasing GSH levels is associated with improved prognosis or protection against disease. Increasing GSH levels is certainly possible, but there is little clinical evidence to support the impressive promises that theory and experimental research have made. In other words, it is not yet fully established whether increasing GSH levels is beneficial, as there are still no solid clinical trials to support this, and there are doubts about the efficacy of treatments in humans. It is not the aim of this review to investigate the reasons for the failures or uncertainties in this area, but we have limited ourselves to assessing which molecules might actually be able to increase GSH levels and by what mechanism, as some are likely to be much more efficient than others.

## Figures and Tables

**Figure 1 antioxidants-12-01094-f001:**
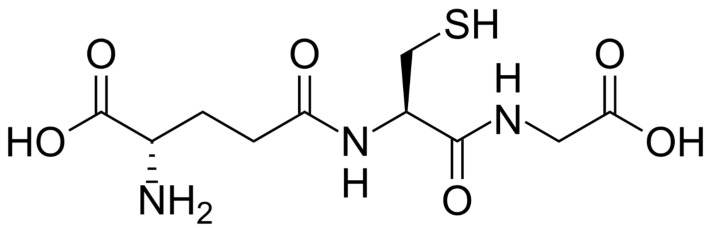
Chemical structure of glutathione.

**Figure 2 antioxidants-12-01094-f002:**
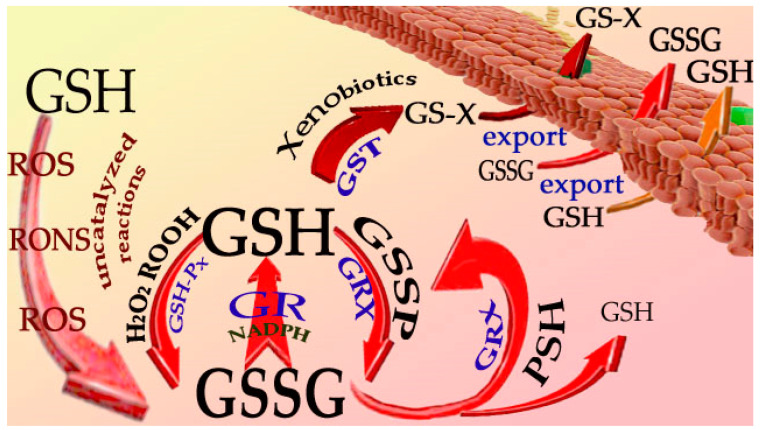
Schematic representations of GSH reactions. Within cells, GSH reacts with oxidants either spontaneously or catalyzed by glutathione peroxidase (GSH-Px). Glutathione disulfide (GSSG) is reduced back to GSH by glutathione reductase (GR). Trans-sulfuration reactions are catalyzed by glutaredoxin (GRX), whereas conjugation with xenobiotics requires catalysis by glutathione transferase (GST). Export of GSH conjugates (GS–X), GSSG and GSH, depends on the activity of multi-drug resistance proteins. ROS, reactive oxygen species; RONS, reactive oxygen/nitrogen species; H_2_O_2_, hydrogen peroxide; ROOH, organic peroxides; GSSP, S-glutathionylated proteins; PSH, protein SH groups.

**Figure 3 antioxidants-12-01094-f003:**
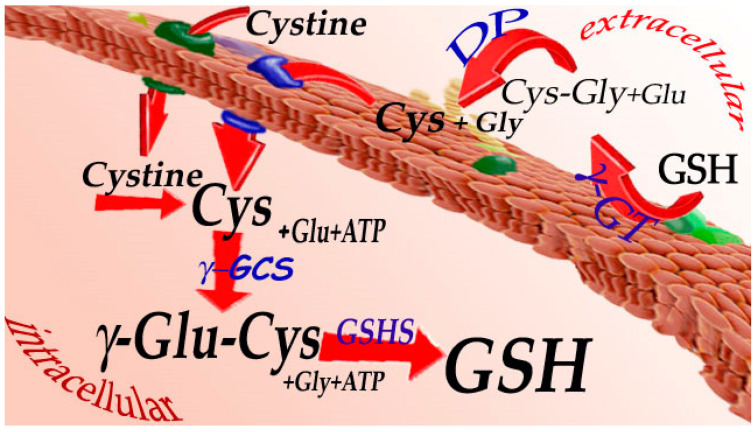
GSH synthesis and degradation. GSH is synthesized by a two-step enzymatic reaction involving γ-glutamylcysteine syntethase (γ-GCS) and GSH synthetases (GSHS) in sequence. γ-GCS catalyzes the formation of the dipeptide γ-glutamylcysteine (γ-Glu-Cys), and GSHS catalyzes the binding of glycine to γ-Glu-Cys to form GSH. This reaction occurs in the cytoplasm of each cell. Cysteine (Cys), which is the rate-limiting factor for this synthesis, can also be obtained by reduction of cystine once it enters the cells. GSH is enzymatically degraded to cysteinylglycine (Cys-Gly) by γ-glutamyltranspeptidase (γ-GT), which is located on the outer surface of plasma membranes. Cys-Gly can then be hydrolyzed into the individual amino acids by extracellular dipeptidases (DP). Glu, glutamic acid; Gly, glycine.

**Table 2 antioxidants-12-01094-t002:** In vivo treatments with different GSH enhancers.

Molecule	Treatment	Species	Mechanism	Reference
GSH	Liposomal	Human	Direct	[66]
GSH	Intranasal	Human	Direct	[67]
GSH	Endovenous	Human	Direct	[67]
GSH	Oral	Human	Direct	[67]
γ-Glu-Cys	Oral	Human	Enzymatic synthesis	[68]
NAC	Oral	Human	Cys delivery	[69]
OTC	Oral	Human	Cys indirect formation	[70]
Green tea	Oral	Human	GSH sparing? Up-regulation antioxidant enzymes?	[71]
DASH diet	Oral	Human	Unknown	[72]
GSH mono ester	Oral	Mouse	Enzymatic release	[73]
γ-Glu-Cys	Diet	Mouse	Enzymatic synthesis	[74]
NACET	Oral	Mouse	NAC and Cys delivery	[75]
S-ethyl cysteine	Oral	Mouse	Cys delivery	[76]
S-methyl cysteine	Oral	Mouse	Cys delivery	[76]
S-propyl cysteine	Oral	Mouse	Cys delivery	[76]
Fumaric acid	Oral	Mouse	Nrf2 inducer	[77]
Taurine	Oral	Mouse	Cys sparing	[78]
Silymarin	Oral	Rat	Antioxidant/Nrf2 inducer/cysteine sparing	[79]
NACET	Oral	Rat	NAC and Cys delivery	[80]
ADT	Oral	Rat	γ-GT inhibition? Nrf2 inducer?	[81]
Bardoxolone	Oral	Monkey	Nrf2 inducer	[82]

Abbreviations: GSH, glutathione, γ-Glu-Cys, γ-glutamylcysteine; Cys, cysteine; NAC, N-acetylcysteine; OTC, 2-oxothiazolidine-4-carboxylate, DASH diet, Dietary Approaches to Stop Hypertension; NACET, N-acetlycysteine ethyl ester; ADT, anethole dithiolthione.

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
