# Peer review of "How to Increase Cellular Glutathione"

_antioxidants, 2023, doi:10.3390/antiox12051094_

Round 1

Reviewer 1 Report

The manuscript entitled „How to increase cellular glutathione” is interesting and includes a valuable summary of the knowledge in the field of glutathione's biological role.  The subject matter may be of interest to other researchers.

I have some minor comments:

Figure 2 – the quality should be improved. I suggest changing the font to get a more professional appearance. All abbreviations from the scheme should be explained in the figure caption (e.g. PSH).

Figure 3 – the quality should be improved. The same as above, I suggest changing the font to get a more professional appearance.

LINE 185 – 5.1. instead of „a”

Line 210, 221, 228, 241 – hyphen should be removed

The manuscript requires editorial proofreading

Author Response

The manuscript entitled „How to increase cellular glutathione” is interesting and includes a valuable summary of the knowledge in the field of glutathione's biological role.  The subject matter may be of interest to other researchers.

I have some minor comments:

Figure 2 – the quality should be improved. I suggest changing the font to get a more professional appearance. All abbreviations from the scheme should be explained in the figure caption (e.g. PSH).

Figure 2 has been modified according to the reviewer’ suggestion. The abbreviations have been explained in the figure caption.

Figure 3 – the quality should be improved. The same as above, I suggest changing the font to get a more professional appearance.

Figure 2 has been modified according to the reviewer’ suggestion.

LINE 185 – 5.1. instead of „a”

We have modified the text accordingly.

Line 210, 221, 228, 241 – hyphen should be removed.

We have modified the text accordingly.

Reviewer 2 Report

The article in its current form is not suitable for publication in a reputable journal, some sections should be remodeled and shortened

1. The abstract lacks information on searching for data (period, database, keywords)

2. the type of review the article is not described. the purpose of the review is also unclear

3. What is new in the review that has not been described in other works:

-Diaz-Vivancos P, de Simone A, Kiddle G, Foyer CH. Glutathione--linking cell proliferation to oxidative stress. Free Radic Biol Med. 2015 Dec;89:1154-64. doi: 10.1016/j.freeradbiomed.2015.09.023

-Sekhar RV. GlyNAC Supplementation Improves Glutathione Deficiency, Oxidative Stress, Mitochondrial Dysfunction, Inflammation, Aging Hallmarks, Metabolic Defects, Muscle Strength, Cognitive Decline, and Body Composition: Implications for Healthy Aging. J Nutr. 2021 Dec 3;151(12):3606-3616. doi: 10.1093/jn/nxab309.

-Kinoshita C, Kubota N, Aoyama K. Glutathione Depletion and MicroRNA Dysregulation in Multiple System Atrophy: A Review. Int J Mol Sci. 2022 Dec 1;23(23):15076. doi: 10.3390/ijms23231507

4. the same remarks also apply to the rest of the manuscript

5. introduction

textbook information on the construction of GSH should be removed and heavily shortened

6. equations of chemical reactions are valuable but not suitable for publication of the article

7. if the drawings were made in a licensed program, they should contain such information in the legend (program name, access)

8. Chapters 2 and 3 should be merged and shortened

9. In both Tables 1 and 2, animal and human studies should be separated

10. I propose to replace the numbering 5a, 5b and 5c with 5.1 and 5.2, and list individual diseases as 5.2.1-6

11. same note to chapter 6

12. The section on regulation lacks information about riboflavin and its contribution to glutathione recycling

SzczukoM, Ziętek M, Kulpa D, Seidler T. Riboflavin - properties, occurrence and its use in medicine. Pteridines 2019; 30:33–47. https://doi.org/10.1515/pteridines-2019-0004

13. Failure to consider relevant scientific articles may lead to misjudgment in cocclusions

Aaseth J, Dusek P, Roos PM. Prevention of progression in Parkinson's disease.

biometals. 2018 Oct;31(5):737-747. doi: 10.1007/s10534-018-0131-5.

14. after specifying the purpose of this review, the method of searching for articles in databases should be described

Minor editing of English language required

Author Response

Comments and Suggestions for Authors

The article in its current form is not suitable for publication in a reputable journal, some sections should be remodeled and shortened

  1. The abstract lacks information on searching for data (period, database, keywords).

The information requested by the reviewer is necessary for systematic reviews or meta-analyzes. Our article is only a review paper, so we selected and commented the most relevant articles in the field according to our expertise. Note that we have worked with some new molecules that are able to increase GSH, e.g., NACET and ADT.

  1. the type of review the article is not described. the purpose of the review is also unclear

We have added a new paragraph in the Introduction where the purpose of the review is indicated.

  1. What is new in the review that has not been described in other works:

-Diaz-Vivancos P, de Simone A, Kiddle G, Foyer CH. Glutathione--linking cell proliferation to oxidative stress. Free Radic Biol Med. 2015 Dec;89:1154-64. doi: 10.1016/j.freeradbiomed.2015.09.023

-Sekhar RV. GlyNAC Supplementation Improves Glutathione Deficiency, Oxidative Stress, Mitochondrial Dysfunction, Inflammation, Aging Hallmarks, Metabolic Defects, Muscle Strength, Cognitive Decline, and Body Composition: Implications for Healthy Aging. J Nutr. 2021 Dec 3;151(12):3606-3616. doi: 10.1093/jn/nxab309.

-Kinoshita C, Kubota N, Aoyama K. Glutathione Depletion and MicroRNA Dysregulation in Multiple System Atrophy: A Review. Int J Mol Sci. 2022 Dec 1;23(23):15076. doi: 10.3390/ijms23231507

We have modified the text accordingly.

  1. the same remarks also apply to the rest of the manuscript

Done.

  1. introduction

textbook information on the construction of GSH should be removed and heavily shortened.

We have shortened this part.

  1. equations of chemical reactions are valuable but not suitable for publication of the article

Since most readers are not experts in this field, some general chemical reactions of GSH might be helpful.

  1. if the drawings were made in a licensed program, they should contain such information in the legend (program name, access)

Done

  1. Chapters 2 and 3 should be merged and shortened

We have shortened and merged the two chapters.

  1. In both Tables 1 and 2, animal and human studies should be separated.

We have separated the animal and human studies in both Tables 1 and 2.

  1. I propose to replace the numbering 5a, 5b and 5c with 5.1 and 5.2, and list individual diseases as 5.2.1-6.

The text has been modified accordingly.

  1. same note to chapter 6

The text has been modified accordingly.

  1. The section on regulation lacks information about riboflavin and its contribution to glutathione recycling

SzczukoM, Ziętek M, Kulpa D, Seidler T. Riboflavin - properties, occurrence and its use in medicine. Pteridines 2019; 30:33–47. https://doi.org/10.1515/pteridines-2019-0004

We have described the contribution of riboflavin to glutathione recycling in the revised text.

  1. Failure to consider relevant scientific articles may lead to misjudgment in cocclusions

Aaseth J, Dusek P, Roos PM. Prevention of progression in Parkinson's disease.

biometals. 2018 Oct;31(5):737-747. doi: 10.1007/s10534-018-0131-5.

The text has been modified accordingly.

  1. after specifying the purpose of this review, the method of searching for articles in databases should be described.

As above stated, the information requested by the reviewer is necessary for the systematic reviews or meta-analysis.

Reviewer 3 Report

1) The manuscript is quite difficult to read because authors use both "authentic" and "authors' original" abbreviations. 

CySS: Cystine or CysS-SCys, NAC: Ac-(stereo)-Cys-OH, NACET: Ac-(stereo)-Cys-OEt, NACA: Ac-(stereo)-Cys-NH2

Furthermore, authors also confused to regulate the abbreviations themselves. Lane 330 what is "gamma-GCL"? 

Authors have to recheck abbreviations in the manuscript  very carefully.

2) Figure 2 and 3

All items (compounds and enzymes) with abbreviations in the figures must be explained in the figure legends. It is quite difficult to find these items (abbreviations) in the text body.

3) lane 442 " N-formyl-Lcysteine" : If the authors keen to describe stereochemistry of cysteine, the other cysteine derivatives will be also describe the stereochemistry "DL", "L" or "D".

4) Journal Abbreviation will be checked for ref 59, 67 and 97  

Minor editing of English language required

Author Response

1) The manuscript is quite difficult to read because authors use both "authentic" and "authors' original" abbreviations. 

CySS: Cystine or CysS-SCys, NAC: Ac-(stereo)-Cys-OH, NACET: Ac-(stereo)-Cys-OEt, NACA: Ac-(stereo)-Cys-NH2

We have checked and standardized the abbreviations.

Furthermore, authors also confused to regulate the abbreviations themselves. Lane 330 what is "gamma-GCL"? 

We meant γ-glutamylcysteine synthase (γ-GCS). We have corrected the text.

Authors have to recheck abbreviations in the manuscript very carefully.

The abbreviations have been rechecked.

2) Figure 2 and 3

All items (compounds and enzymes) with abbreviations in the figures must be explained in the figure legends. It is quite difficult to find these items (abbreviations) in the text body.

We have added the explanation of the abbreviations for Figure 2 and 3 in the revised text.

3) lane 442 " N-formyl-Lcysteine" : If the authors keen to describe stereochemistry of cysteine, the other cysteine derivatives will be also describe the stereochemistry "DL", "L" or "D".

We have removed the indication of the stereoisomers of cysteine.

4) Journal Abbreviation will be checked for ref 59, 67 and 97.

Checked.  

Reviewer 4 Report

Due to its high intracellular concentration, widespread presence, and ability to react with electrophiles of the sulfhydryl group in its cysteine component, Glutathione (GSH) possesses unique antioxidant properties. In numerous conditions where oxidative stress is believed to contribute to the disease process, a decrease in GSH concentration is often observed, making cells more vulnerable to oxidative damage. Consequently, there is a growing interest in identifying effective methods for elevating cellular glutathione levels, both for disease prevention and treatment purposes. This review examines various strategies for enhancing cellular GSH reserves, including GSH and its derivatives, NRf-2 activators, cysteine prodrugs, certain foods, and specialized diets. The mechanisms by which these compounds can function as GSH enhancers, as well as the related pharmacokinetic concerns, advantages, and disadvantages, are also discussed.

As a reviewer, I would like to provide the following critical review report for the paper titled “How to increase cellular glutathione”.

1.      The introduction provides a good overview of the properties and functions of GSH, but it would benefit from a clearer statement of the paper’s aims and objectives.

2.      The role of GSH in radical scavenging and detoxification is well explained, but the section could be improved by linking these functions to the importance of maintaining cellular GSH levels.

3.      The section on factors that influence intracellular GSH levels is well researched and informative, but it could be improved by clarifying the role of each factor in maintaining cellular GSH levels.

4.      The section on GSSG concentrations could be improved by providing a clearer explanation of the factors that can cause artificially elevated GSSG levels.

5.      The paper could benefit from a clearer discussion of the relevance of maintaining cellular GSH levels in different disease states and the potential clinical applications of GSH boosters.

6.      The mechanism of action of whey protein and glutathione and their effects on increasing cellular glutathione levels mentioned in this paragraph was not further explained. It is recommended to add related explanations to help readers better understand.

7.      The authors should discuss the severe limitations of their approach and give future directions for research in the field of GSH research.

Author Response

Due to its high intracellular concentration, widespread presence, and ability to react with electrophiles of the sulfhydryl group in its cysteine component, Glutathione (GSH) possesses unique antioxidant properties. In numerous conditions where oxidative stress is believed to contribute to the disease process, a decrease in GSH concentration is often observed, making cells more vulnerable to oxidative damage. Consequently, there is a growing interest in identifying effective methods for elevating cellular glutathione levels, both for disease prevention and treatment purposes. This review examines various strategies for enhancing cellular GSH reserves, including GSH and its derivatives, NRf-2 activators, cysteine prodrugs, certain foods, and specialized diets. The mechanisms by which these compounds can function as GSH enhancers, as well as the related pharmacokinetic concerns, advantages, and disadvantages, are also discussed.

As a reviewer, I would like to provide the following critical review report for the paper titled “How to increase cellular glutathione”.

  1. The introduction provides a good overview of the properties and functions of GSH, but it would benefit from a clearer statement of the paper’s aims and objectives.

We have modified the Introduction and added a new statement on the paper’s aims and objectives.

  1. The role of GSH in radical scavenging and detoxification is well explained, but the section could be improved by linking these functions to the importance of maintaining cellular GSH levels.

This section has been improved as suggested in the chapter: “Reactions of GSH”.

  1. The section on factors that influence intracellular GSH levels is well researched and informative, but it could be improved by clarifying the role of each factor in maintaining cellular GSH levels.

We agree, and a brief summary of this can be found in the introduction. However, we do not think that further improvement and development of this point is absolutely necessary, since we want to direct the readers' attention to the crucial question: "which molecules are able to increase GSH?"

  1. The section on GSSG concentrations could be improved by providing a clearer explanation of the factors that can cause artificially elevated GSSG levels.

Done.

  1. The paper could benefit from a clearer discussion of the relevance of maintaining cellular GSH levels in different disease states and the potential clinical applications of GSH boosters.

Even if this goes beyond the aim of the review, we tried to clarify this point in the conclusions.

  1. The mechanism of action of whey protein and glutathione and their effects on increasing cellular glutathione levels mentioned in this paragraph was not further explained. It is recommended to add related explanations to help readers better understand.

This paragraph hast been improved as suggested.

  1. The authors should discuss the severe limitations of their approach and give future directions for research in the field of GSH research.

We limited our analysis to molecules reported to increase GSH. This represents both a strength and a weakness of our work. The strength is that we believe that it needs to be clarified which molecules are really able to increase GSH and to what extent, the weakness is that we do not know the effect of increased GSH because there are no robust clinical data in the literature. This aspect should be the focus of future research on GSH enhancers. This point has been better clarified in both the introduction and conclusion of the revised article.

Round 2

Reviewer 3 Report

The revised manuscript meets the quality for publication.